# Osteoporosis Treatment with Anti-Sclerostin Antibodies—Mechanisms of Action and Clinical Application

**DOI:** 10.3390/jcm10040787

**Published:** 2021-02-16

**Authors:** Martina Rauner, Hanna Taipaleenmäki, Elena Tsourdi, Elizabeth M. Winter

**Affiliations:** 1Divisions of Endocrinology and Molecular Bone Biology, Department of Medicine III, Medical Center, Technische Universität Dresden, 01307 Dresden, Germany; elena.tsourdi@ukdd.de; 2Molecular Skeletal Biology Laboratory, Department of Trauma and Orthopedic Surgery, University Medical Center Hamburg-Eppendorf, 20246 Hamburg, Germany; h.taipaleenmaeki@uke.de; 3Division of Endocrinology, Center for Bone Quality, Department of Medicine, Leiden University Medical Center, 2300 RC Leiden, The Netherlands; E.M.Winter@lumc.nl

**Keywords:** sclerostin, wnt signaling, osteoblast, osteoclast, osteoporosis, romosozumab

## Abstract

Osteoporosis is characterized by reduced bone mass and disruption of bone architecture, resulting in increased risk of fragility fractures and significant long-term disability. Although both anti-resorptive treatments and osteoanabolic drugs, such as parathyroid hormone analogues, are effective in fracture prevention, limitations exist due to lack of compliance or contraindications to these drugs. Thus, there is a need for novel potent therapies, especially for patients at high fracture risk. Romosozumab is a monoclonal antibody against sclerostin with a dual mode of action. It enhances bone formation and simultaneously suppresses bone resorption, resulting in a large anabolic window. In this opinion-based narrative review, we highlight the role of sclerostin as a critical regulator of bone mass and present human diseases of sclerostin deficiency as well as preclinical models of genetically modified sclerostin expression, which led to the development of anti-sclerostin antibodies. We review clinical studies of romosozumab in terms of bone mass accrual and anti-fracture activity in the setting of postmenopausal and male osteoporosis, present sequential treatment regimens, and discuss its safety profile and possible limitations in its use. Moreover, an outlook comprising future translational applications of anti-sclerostin antibodies in diseases other than osteoporosis is given, highlighting the clinical significance and future scopes of Wnt signaling in these settings.

## 1. Introduction

Osteoporosis is a systemic disease that is characterized by low bone mass, microarchitectural deterioration, and impaired bone quality predisposing individuals to an increased risk of fracture [1]. Osteoporosis is the most common bone disease, and its prevalence increases with age. However, not only age is a significant risk factor for osteoporosis, but also sex. With declining concentrations of the bone-protective hormone estrogen after menopause, women are particularly susceptible to developing osteoporosis [2]. In fact, about 20% of women aged over 65 are affected by osteoporosis in the European Union as compared to about 7% of men [3]. At the age of 80, these numbers are twice as high, affecting ~50% of women and 17% of men [3]. Thus, as osteoporotic fractures, which most commonly occur at the hip, spine, humerus, and forearm, are common and associated with significant morbidity and a 20% mortality rate one year post-fracture [4], prevention of fractures is key, not only to prevent pain and long-term disability for affected individuals, but also to reduce socioeconomic costs. Importantly, patient adherence is pivotal to drugs’ anti-fracture efficacy, as has been demonstrated by less frequently administered subcutaneous regimens [5,6].

One effective measure to prevent fractures is to enhance bone mass and quality. As bone is a highly dynamic tissue that undergoes constant remodeling to adapt to changing functional and metabolic demands and to repair micro-damages that occur throughout life, there is the possibility to harness this natural process for therapeutic purposes. The bone remodeling cycle starts when osteoclast precursors are attracted to a site of future bone remodeling through secreted products by osteocytes and other cells in the vicinity of damaged bone. Once fully differentiated, osteoclasts resorb bone. Within the reversal phase, osteoclasts retract and make space for osteoblasts to refill the resorbed area. Once they have completed bone formation, they either become embedded into the bone matrix as osteocytes or they remain in a quiescent state at the bone surface as bone lining cells [7]. Ideally, the amount of newly formed bone equals the amount of degraded bone. However, in osteoporosis, bone resorption exceeds bone formation, leading to loss of bone mass.

Most osteoporosis therapies aim at inhibiting osteoclastic bone resorption, while only a few are capable of actively promoting the generation of new bone tissue. By neutralizing the actions of the Wnt inhibitor sclerostin, romosozumab is one of those few osteo-anabolic therapies. Romosozumab stimulates bone formation by osteoblasts and at the same time inhibits bone resorption by osteoclasts, leading to a large anabolic window. In this narrative review, we will revisit the actions of sclerostin on bone remodeling as well as extra-skeletal tissues and describe the newest advances in the application of romosozumab in the treatment of osteoporosis and other bone diseases in terms of efficacy, sequential therapy, and side effect profile.

## 2. Sclerostin

### 2.1. Expression, Mechanisms of Action, and Regulation of Sclerostin in Bone

Sclerostin (*SOST*) is a secreted monomeric protein of 24 kDa that is encoded by the *SOST* gene on chromosome 17q12–21 and is a major inhibitor of bone formation [8,9]. It inhibits osteoblast differentiation by binding to its receptors low density lipoprotein receptor-related protein 5 (LRP5) and low-density lipoprotein receptor-related protein 6 (LRP6), thus impeding downstream activation of canonical Wnt signaling [10,11,12]. Low density lipoprotein receptor-related protein 4 (LRP4) can also associate with sclerostin, however, unlike LRP5/6, LRP4 enhances the suppressive effect of sclerostin on Wnt signaling [13]. Sclerostin-mediated inhibition of bone morphogenetic protein (BMP) signaling has also been demonstrated in osteoblasts and osteocytes, indicating that crosstalk between Wnt and BMP signaling may be relevant [8,14,15]. Nonetheless, its inhibitory effect on Wnt signaling appears to be the predominant effect of sclerostin. Besides blocking osteoblast differentiation, sclerostin induces osteoblast apoptosis, thereby limiting the amount of bone matrix that can be produced [16]. Via increasing the receptor activator of NFκB ligand (RANKL): osteoprotegerin ratio in osteocytes, sclerostin has been also shown to stimulate osteoclastogenesis [17]. In line with this, suppression of bone resorption has been identified as a crucial mechanism of bone accrual in organisms lacking sclerostin [9,18] and in rodents, monkeys, and humans after inhibition of sclerostin using antibodies (Figure 1) [19,20].

Sclerostin is highest expressed in osteocytes, but has also been detected in various other tissues including arteries, calcifying smooth muscle cells, cartilage, kidney, adipose tissue, bile ducts, epididymis, and part of the cerebellum [21,22,23,24,25,26]. In osteogenic cells, which have been mostly studied thus far regarding the regulation of sclerostin, *SOST* expression is activated at its proximal promoter by Runx2 and osterix, two main osteoblastic transcription factors [27,28]. Moreover, Mef2c, BMP signaling, and methylation of the proximal promoter induce *SOST* expression, while IRF7 has been shown to reduce its expression [9,29,30,31] (for more details on transcriptional regulation see [32]). At a systemic level, sclerostin is inhibited most prominently by parathyroid hormone (PTH) and mechanical loading, which both exert their osteoanabolic actions through downregulating sclerostin (Figure 1) [33,34,35,36]. While suppression of *SOST* expression after PTH treatment is dependent on the cAMP/PKA signaling pathway, transforming growth factor-β (TGFβ) signaling has been shown to be a key component in the suppression of *SOST* by mechanical loading [37,38,39]. Various hormones including vitamin D, glucocorticoids, estrogen, and androgens have been shown to induce *SOST* expression, although data on most of them are quite discrepant [32]. More consistent data are available for the pro-inflammatory cytokines tumor necrosis factor-α (TNFα) and interleukin-1β [40,41,42,43], as well as hypoxia signaling via hypoxia-inducible factor-1α [44,45,46], which induces *SOST* expression (Figure 1). Cytokines of the interleukin-6 superfamily including oncostatin M, cardiotrophin-1, and leukemia inhibitory factor inhibit *SOST* expression.

### 2.2. Sclerostin Is a Critical Regulator of Bone Mass

Mutations that decrease *SOST* expression lead to disorders of generalized skeletal hyperostosis. In humans, two rare autosomal recessive diseases have been described including sclerosteosis (OMIM #269500), which results from mutations in the coding region of the *SOST* gene, and van Buchem’s disease (OMIM #239100, also known as *hyperostosis corticalis generalisata*), caused by a mutation in the downstream enhancer region ECR5 (evolutionary conserved region 5) of the same gene [47,48,49]. Both diseases are characterized by high bone mass, albeit to a somewhat lesser extent in van Buchem’s disease. Besides high bone mass, patients with sclerosteosis frequently present with excessive height and weight, syndactyly, and progressive skeletal overgrowth, especially in the skull and face, resulting in distinctive facial features (e.g., mandibular overgrowth and frontal bossing), cranial nerve entrapment causing facial palsy and deafness, and potentially lethal elevation of intracranial pressure [49]. Patients with heterozygous mutations exhibit moderately elevated bone mass without any other clinical manifestations. Similarly to humans, mice lacking *SOST* also recapitulate the high bone mass phenotype due to enhanced osteogenesis [50]. In line, overexpression of *SOST* leads to low bone mass due to inhibited bone formation [8]. Besides regulating bone mass, sclerostin has also been shown to regulate the composition of the bone matrix. Bone from mice and humans with *SOST* deficiency shows a lower matrix mineralization and a higher relative proteoglycan content, which may contribute to the increased bone strength [51]. 

## 3. Lessons Learnt about the Mode of Action of Sclerostin from Animal Models 

### 3.1. Genetic Animal Models of Sclerostin Deficiency or Overexpression

While the aforementioned human diseases provided important evidence for sclerostin as a critical regulator of bone anabolic signaling, generation of a sclerostin knockout mouse model has allowed further investigation of the pathway [50]. Deletion of *SOST* using gene targeting resulted in a high bone mass phenotype similar to individuals with sclerosteosis and van Buchem’s disease. The significant increase of bone mass was due to an increased bone formation of trabecular bone as well as at periosteal and endosteal surfaces of cortical bone. At a cellular level, absence of sclerostin resulted in an increased osteoblast surface, while osteoclast surface was not affected. The uncoupling between bone formation and resorption was supported by increased concentrations of serum osteocalcin as a surrogate marker for osteoblast activity, while serum levels of osteoclast marker tartrate-resistant acid phosphatase-5b (TRACP-5b) remained unchanged. Because of an enhanced mineral apposition and increased lamellar bone, *SOST*-deficient mice exhibited increased bone strength in the lumbar vertebrae and femur in both female and male counterparts [50]. Conversely, transgenic mice expressing human *SOST* from the mouse osteocalcin promoter had reduced trabecular bone mass, thinner cortices, impaired lamellar bone formation, and chondrodysplasia [8]. At a tissue level, histomorphometric analyses revealed that these transgenic mice depicted a decreased osteoblast surface and a reduced bone formation rate compared to the littermate controls, while bone resorption parameters were not changed. Consequently, vertebrae and femurs of *SOST* transgenic mice were fragile and more prone to fractures, as demonstrated by compression and four-point bending tests [8]. Consistent results were also reported by Loots and colleagues who generated a transgenic mouse expressing human *SOST* from a bacterial artificial chromosome [52]. Compared to controls, the *SOST* transgenic mice had reduced bone mineral density (BMD) and bone mass accompanied by a decreased bone formation rate and limb deformities [52], further supporting the inhibitory role of sclerostin in bone formation. 

Both sclerostin-deficient and overexpressing mouse models have been instrumental in understanding the role of sclerostin in mediating the bone response to mechanical loading. Mechanical loading promotes bone formation, while reduced mechanical strain leads to bone loss. Interestingly, sclerostin expression was shown to decrease under enhanced mechanical loading conditions in an ulnar loading model (Figure 1) [34]. Conversely, during hindlimb unloading, *SOST* expression was markedly increased, indicating that loading-induced changes in bone mass might depend on sclerostin. Indeed, sclerostin-deficient mice were resistant to bone loss caused by mechanical unloading [35], whereas mice expressing sclerostin under the Dentin matrix protein 1 (Dmp1) had an impaired anabolic response to loading [36]. Together, these studies demonstrated that downregulation of *SOST* in osteocytes is required in the mechanotransduction cascade that activates bone formation. 

The strong bone phenotype observed in genetic mouse models together with the high bone mass in patients deficient of sclerostin indicated that sclerostin might be a novel therapeutic target and encouraged the development of specific antibodies against sclerostin as a bone anabolic approach to treat osteoporosis and other bone loss conditions. 

### 3.2. Preclinical Models of Sclerostin Inhibition

As part of the preclinical development program of anti-sclerostin antibodies (Scl-Ab), several animal models were used. The first efficacy studies were performed in ovariectomized mice and rats as an animal model of postmenopausal osteoporosis. Unlike rodents, monkeys have a very similar bone remodeling process as humans, and thus female cynomolgus monkeys serve as a non-human primate model in accordance with regulatory guidance on the Evaluation of Medicinal Products in the Treatment of Primary Osteoporosis [53]. Together, the animal studies outlined in this chapter have demonstrated that sclerostin antibody treatment increases bone formation as well as bone mass and strength in aged ovariectomized rats and monkeys. In addition, animal studies have been conducted to investigate the effect of treatment discontinuation, transition to antiresorptive therapy, and the effect of treatment on bone matrix composition. 

The first evidence suggesting that antibody-mediated inhibition of sclerostin increases bone mass was obtained in ovariectomized osteopenic rats that were treated with a monoclonal Scl-Ab for five weeks. Treatment with Scl-Ab strongly increased bone formation of trabecular, periosteal, endocortical, and intracortical surfaces, leading to greater bone mass and strength compared to non-ovariectomized control rats at the lumbar spine and femurs [54]. These effects are gender-independent as a similar phenotype was observed in gonad-intact aged male rats treated with Scl-Ab for five weeks [55]. A more detailed understanding of the effects of Scl-Ab on bone remodeling in various skeletal compartments was obtained from a study comparing skeletal responses after six- and 26-week Scl-Ab treatment of ovariectomized female rats [56]. Cumulative effects on bone mass and strength were observed at multiple skeletal sites after six and 26 weeks of treatment. However, bone formation parameters after six and 26 weeks varied depending on the skeletal site and compartmental surface. While the six-week treatment increased trabecular, endocortical, and periosteal bone formation [54,56], response to 26-week treatment was not only different between lumbar vertebra and proximal tibia, but also between the trabecular and endocortical versus periosteal surfaces. Specifically, bone formation remained stimulated on the endocortical surfaces but returned to the level of vehicle treated controls on periosteal sites. The strong, cumulative increase in bone mass could be explained by a greatly increased bone formation at the lumbar spine and proximal tibia throughout the 26-week treatment with Scl-Ab [56]. Neither pretreatment nor co-treatment with the antiresorptive agent alendronate affected Scl-Ab-induced increase in bone formation and bone mass [57]. Together, these studies showed that Scl-Ab increases bone mass and improved bone quality in rodents, providing a base to investigate the effect of sclerostin inhibition in primates.

Non-human primates have similarities in bone remodeling, reproductive physiology, and immune system with humans and are therefore a suitable model to evaluate promising drugs affecting the skeleton [53]. The effect of Scl-Ab on BMD and bone turnover was first investigated in adolescent gonad-intact female monkeys using three doses (3, 10, or 30 mg/kg) of humanized Scl-Ab once a month for two months [58]. Similar to rodents, Scl-Ab treatment significantly increased dose-dependent BMD at the femoral neck, radius, and tibial metaphysis. Furthermore, bone strength was increased in monkeys treated with 30 mg/kg of Scl-Ab. Detailed analysis of serum bone turnover markers revealed a dose-dependent increase of bone formation markers serum type 1 aminoterminal propeptide (P1NP) and osteocalcin and a temporal reduction in bone resorption marker serum C-telopeptide (CTX). A strong increase in bone formation pertaining to the periosteal, trabecular, and endocortical surfaces with administration of the highest dose of Scl-Ab was confirmed by histomorphometry [59]. Interestingly, indices of bone resorption were unchanged, indicating that Scl-Ab-mediated bone formation is uncoupled from bone resorption. Besides physiological bone remodeling, cynomolgus monkeys respond to estrogen withdrawal with bone loss recapitulating postmenopausal osteoporosis occurring in women. Although bone loss does not progress to osteoporosis but rather osteopenia, ovariectomized cynomolgus monkeys are a frequently used model in preclinical studies of testing the efficacy and safety of osteoporosis drugs, including Scl-Ab [53]. A monthly treatment of female monkeys four months after ovariectomy with two doses of Scl-Ab (3 and 30 mg/kg) dose-dependently increased trabecular BMD at the lumbar spine and femur neck after 12 months [59], corresponding to a significant increase in bone strength at lumbar bodies with both doses and at the femur with the 30 mg/kg dose. Consistent with previous studies, Scl-Ab caused a rapid increase in bone formation. Interestingly, when treatment was discontinued after six months followed by six-month treatment with vehicle, BMD gradually decreased to baseline, while bone strength and material properties were partially maintained especially at cortical sites [59]. Treatment of monkeys with Scl-Ab did not affect their general health indicating that pharmacological inhibition of sclerostin is an efficient and safe treatment to increase bone mass in primates.

More detailed investigation of the mechanism of action of Scl-Ab at tissue level was performed in male cynomolgus monkeys treated with Scl-Ab for 10 and 28 weeks [60]. Kinetic reconstruction based on sequential fluorochrome labels revealed a transient increase of mineral apposition rate in remodeling sites at three weeks after treatment. Histological analysis of modeling and remodeling bone formation sites in the vertebral cancellous bone showed a period of modeling-based bone formation at the early phase of treatment at 10 weeks. This was accompanied by an increased wall thickness at remodeling sites and reduced remodeling space due to decreased resorption [60]. Consistent with other reports, during long-term treatment, the effect of modeling-based bone formation attenuated, while bone resorption remained suppressed, resulting in a positive remodeling balance. Scl-Ab treatment also alters vertebral trabecular bone microarchitecture, as demonstrated by three-dimensional dynamic histomorphometry [61]. Upon Scl-Ab treatment, rod-like trabeculae were converted to plate-like trabeculae at a higher rate than in vehicle treated male monkeys, which was accompanied by an improved mechanical performance [61].

Together, these preclinical studies demonstrate that Scl-Ab treatment induces a rapid but transient increase in bone formation independent of skeletal site and surface. During the treatment period, bone formation attenuates, while bone resorption remains suppressed leading to a progressive increase in BMD. This dual bone anabolic and antiresorptive mode of action uncouples bone formation and resorption with a net gain in bone mass, thus representing a potent osteoporosis drug.

## 4. Monoclonal Antibodies against Sclerostin in Human Osteoporosis Treatment

Romosozumab and blosozumab are monoclonal antibodies that bind to sclerostin, reducing its inhibition of Wnt signaling and bone formation. In the following chapter, their clinical development will be described. 

### 4.1. Blomosozumab

The phase I study with blosozumab, a recombinant humanized antibody, investigated single or multiple escalating doses in the setting of postmenopausal osteoporosis [62]. There were dose-dependent increases in bone formation markers and BMD after single and multiple doses of blosozumab, while bone resorption markers were dose-dependently suppressed. All doses were well-tolerated. Prior bisphosphonate use had no significant impact on the alterations on bone remodeling or bone mass [62]. The phase II study comprised increasing subcutaneous doses of blosozumab (180 mg every four weeks, 180 mg every two weeks, 270 mg every two weeks, or placebo) administered to women with postmenopausal osteoporosis (*n* = 120) with a lumbar spine *T*-score −2.0 to −3.5 [63]. Bone formation markers increased rapidly with a dose-related increase, and there was a considerable reduction of bone resorption, as assessed with CTX during the first two months of treatment, which was not maintained thereafter. Blosozumab treatment resulted in significant dose-related increases in spine, femoral neck, and total hip BMD compared with placebo. In the highest dose group, mean increases in BMD from baseline were 17.7% at the spine and 6.2% at the total hip at 12 months. Thus, this study yielded similar efficacy results to the romosozumab program (see below). Following treatment discontinuation, bone turnover markers remained close to baseline levels, but BMD gradually declined at the one-year follow-up [64]. In detail, lumbar spine BMD remained significantly greater than placebo in women initially treated with the highest blomosozumab dose with more modest residual effects at the total hip [64]. No further trials with blomosozumab, including Phase III trials, were reported or are registered on clinicaltrials.gov as of December 24, 2020.

### 4.2. Romosozumab

#### 4.2.1. Phase I and II Studies: Results on BMD and Bone Turnover Markers 

Romosozumab is a fully humanized and highly specific monoclonal IgG2 antibody against sclerostin, which has progressed into clinical practice. A phase I multidose study included 72 healthy men and postmenopausal women receiving a single dose of romosozumab [65]. This treatment resulted in dose-related increases of bone formation markers up to 200% above baseline and decrease of bone resorption markers of up to 50% below baseline, while a 4–5% increase in lumbar spine BMD was achieved at the maximum dose of romosozumab [65]. These findings were verified by a second multi-dose phase I study in which romosozumab was administered over three months [66].

In a phase II dose-finding study, 419 women with postmenopausal osteoporosis received any one of five romosozumab subcutaneous dosing regimens (ranging from 70 mg every three months to 210 mg every month), placebo, alendronate, or teriparatide for 12 months [67]. Romosozumab treatment resulted in a rapid increase of bone formation markers, which peaked at 1–3 months, returned to baseline by month 6, and remained below baseline until the end of study. Conversely, bone resorption markers rapidly decreased and remained suppressed until month 12. At 12 months, the mean rise in BMD was 11.3% at the lumbar spine and 4.1% at the total hip, with a dose of 210 mg romosozumab per month as compared to placebo (both *p* < 0.001) [67] (Table 1). BMD rise was greater with romosozumab than with the active comparators alendronate or teriparatide [67]. Similar results to the international phase II study [67] were reported in a smaller phase II study in Japanese women with postmenopausal osteoporosis who were randomized to receive placebo or romosozumab (70, 140, or 210 mg) subcutaneously once monthly for 12 months [68]. Based on these results, the highest dosing regimen of 210 mg/month was further developed in the phase III studies.

#### 4.2.2. Phase III Studies: Results on BMD and Anti-Fracture Activity

The first Phase III study was the Fracture Study in Postmenopausal Women with Osteoporosis (FRAME) [69]. This study included 7180 women with postmenopausal osteoporosis who received either subcutaneous injections of 210 mg romosozumab per month or placebo for 12 months, followed by an open-label period of 12 months where all participants were treated with subcutaneous denosumab 60 mg every six months. Denosumab is a monoclonal antibody against RANKL. It suppresses bone resorption by preventing osteoclast differentiation, proliferation, and activation. Due to coupling, osteoblasts and consequently bone formation are suppressed as well. However, there is a net increase in bone mineral density leading to effective fracture prevention [70]. During the first 12 months of the FRAME study, there was a 73% reduction for vertebral fractures (*p* < 0.001) and a 36% reduction for clinical fractures (*p* = 0.008), while no statistically significant difference was reported for non-vertebral fractures. During the second year of the study where all subjects were treated with denosumab, the cumulative risk for vertebral fractures was significantly lower in the former romosozumab users (five patients vs. 25 patients in the former placebo group), translating to a 75% relative risk reduction of new vertebral fractures (*p* < 0.001). In addition, a persistent benefit was reported across other fracture categories (Table 1) [69].

The effect of subcutaneous romosozumab 210 mg per month vs. oral alendronate 70 mg per week over 12 months, followed by further two months where all subjects received alendronate, was investigated in a second Phase III fracture endpoint trial (ARCH) [71]. Over a period of 24 months, a 48% lower risk of new vertebral fractures was reported in the romosozumab-to-alendronate group (*p* < 0.001) (Table 1). Moreover, a 27% lower risk for clinical fractures was seen with romosozumab (*p* < 0.001), while non-vertebral fractures were reduced by 19% (*p* = 0.04), and hip fractures were reduced by 38% (*p* = 0.02) through romosozumab treatment [71].

The BRIDGE study [72] aimed to extrapolate the fracture benefit observed in women with osteoporosis in FRAME [69] to men. As such, the BRIDGE study was a Phase III study, which recruited 245 men with osteoporosis or osteopenia and a history of fragility fracture and randomized subjects to receive either romosozumab 210 mg per month or placebo for 12 months. After 12 months, BMD increased at the lumbar spine by 12.1% in romosozumab treated subjects vs. 1.2% in the placebo group (*p* < 0.001), with more modest albeit significant BMD increases at the total hip and femoral neck (Table 1) [72]. These differences were observed as early as month 6, while evaluations of bone turnover markers revealed a similar profile to past romosozumab studies in postmenopausal women. These findings suggest that romosozumab treatment could also reduce fractures in men with osteoporosis, although an anti-fracture study with romosozumab has not yet been performed in the setting of male osteoporosis. Of note, additional studies are also needed to validate the efficacy of romosozumab in men in the setting of secondary causes of male osteoporosis such as hypogonadism, diabetes mellitus, or primary aldosteronism [73,74,75].

Since in clinical practice, patients pre-treated with bisphosphonates can be switched to osteo-anabolic agents in case of poor response or higher risk fracture than previously estimated, the Phase III STRUCTURE study recruited 436 postmenopausal women treated with alendronate for a mean of 6.2 years before being randomized to receive either teriparatide or romosozumab for the next 12 months [76]. Romosozumab significantly increased BMD at the total hip, lumbar spine, and femoral neck relative to baseline and teriparatide. Of note, romosozumab treatment resulted in BMD gains of 2.6% at the total hip (relative to baseline) through 12 months, which was significantly greater than teriparatide treatment (−0.6% from baseline), leading to a between-group difference of 3.2% (*p* < 0.001) (Table 1) [76]. A more detailed analysis revealed that changes in the cortical bone were directionally opposed with the two agents, with teriparatide causing reduced BMD and bone mineral content in the cortical compartment [76], confirming results of a previous study with teriparatide [77] and indicating that teriparatide, as opposed to romosozumab, appears to have a catabolic effect in cortical bone.

**Table 1 jcm-10-00787-t001:** Romosozumab Phase II/III clinical studies.

Type of Study	Aim	Study Population	Main Findings			Reference
		(Patients)	BMD	Fx	BTMs	
Phase II international, multicenter, placebo-controlled, parallel-group RCT	BMD changes after 12 months of Romo (5 doses) vs. placebo vs. ALN vs. TPTD	Postmenopausal women with T-score (FN/TH/LS) between −2.5 and −3.5 (419)	BMD LS: ↑11.3% (Romo 210 mg/month) ↑7.1% (TPTD) ↑4.1% (ALN) ↓0.1% (placebo)	NR	Romo: P1NP ↑ with maximum after 1 month, then gradual decrease CTX ↓ sustained through month 12	[67]
Phase III international, multicenter, double-blind, placebo-controlled, parallel-group RCT (FRAME)	Primary: VFx at month 12 (Romo vs placebo) and month 24 (all Dmab) Secondary: (i) Clinical Fx, (ii) non-VFx	Postmenopausal women with T-score (FN/TH/LS) between −2.5 and −3.5 (7180)	BMD compared to placebo: ↑ at LS 13.3%; at TH 6.9%; at FN 5.9%	↓ VFx at month 12 (73% lower RR with Romo) and sustained at month 24 ↓ Clinical VFx at month 12 (36% lower RR with Romo) No significant reduction of non-VFx	Romo: P1NP ↑ with maximum after 14 days, then gradual decrease CTX ↓ with minimum after 14 days, sustained decrease through month 12	[69]
Phase III, international, multicenter, double-blind trial RCT vs. active comparator (ARCH)	Primary: (i) VFx at month 24 (0–12 month Romo vs ALN and 12–24 month all ALN) (ii) Clinical Fx at primary analysis Secondary: (i) BMD LS, FN, TH (ii) non-VFx at primary analysis	Postmenopausal women with T-score (FN/TH) ≤ −2.0 and two or more VFx or a hip Fx (4093)	Month 12 BMD LS:↑13.7% (Romo) ↑5.0% (ALN) Month 24 LS:↑15.2% (Romo→ALN) ↑7.1% (ALN→ALN)	↓ VFx at month 24 (48% lower RR with Romo compared to ALN) ↓ Clinical VFx at primary analysis (27% lower RR with Romo compared to ALN)	Romo: P1NP ↑ with maximum after1 month, then gradual decrease and suppression after switch to ALN CTX ↓ sustained suppression through month 24	[71]
Phase III international, multicenter, double-blind, placebo-controlled, parallel-group RCT in men (BRIDGE)	Primary: BMD LS changes after 12 months of Romo vs. placebo Secondary: (i) BMD FN/TH changes after 12 months of Romo vs. placebo (ii) BTMs iii) bone histology (subset of 20)	Men aged 55–90 years, with T-score (FN/TH/LS) between −1.5 and −2.5 and fragility VFx or non-VFx (245)	Month 12 BMD LS:↑12.1% (Romo) ↑1.2% (placebo) TH:↑2.5% (Romo) ↓0.5% (placebo) FN:↑2.2% (Romo) ↓0.2% (placebo)	NR	Romo: P1NP ↑ with maximum after 1 month, then gradual decrease CTX ↓ with minimum after 1 month, sustained decrease through month 12 * histology: sustained reduction in bone resorption	[72]
Phase III, international, multicenter, double-blind trial RCT vs. active comparator (STRUCTURE)	Primary: BMD TH change at 12 months Secondary: (i) BMD TH change at 12 months (ii) cortical BMD at TH at 6 and 12 months (iii) BMC at TH at 6 and 12 months iv) estimated bone strength at TH at 6 and 12 months	Postmenopausal women with T-score (FN/TH/LS) ≤ −2.5 and fragility VFx or non-VFx; ≥ 3 years of prior BP therapy (436)	Month 12 BMD TH:↑2.6% (Romo) ↓0.6% (TPTD) Month 12 cortical BMD TH:↑1.1% (Romo) ↓3.6% (TPTD) Month 12 estimated bone strength TH:↑2.5% (Romo) ↓0.7% (TPTD) Month 12 BMC TH:↑3.6% (Romo) ≈0.0% (TPTD)	NR	NR	[76]

**Abbreviations**: ALN: alendronate; BMC: bone mineral content; BMD: bone mineral density; BP: bisphosphonate; BTMs: bone turnover markers; CTX: C-terminal-cross-linking telopeptide of type 1; Dmab: denosumab; FN: femoral neck; Fx: fractures; LS: lumbar spine; NR: not reported; non-VFx: non-vertebral fracture; P1NP: Procollagen 1 Intact N-Terminal Propeptide; RCT: randomized controlled trial; RR: relative risk; Romo: romosozumab; TH: total hip; TPTD: teriparatide; VFx: vertebral fractures.

#### 4.2.3. Post-Hoc Analyses and Additional Exploratory End-Points

As mentioned above, the FRAME study established the anti-fracture efficacy of romosozumab with regard to the incidence of new vertebral fractures and clinical fractures. In terms of non-vertebral fractures, a non-significant reduction of 25% compared to placebo (*p* = 0.10) was observed. Since a large number (43%) of participants in FRAME were from Latin America, a post-hoc analysis of non-vertebral and clinical fracture incidence was performed in the Latin America and rest-of-world groups [78]. Although the significant anti-fracture effect pertaining to vertebral fractures was confirmed in both the Latin America and the rest-of-world groups, for clinical (mainly comprising non-vertebral fractures) and non-vertebral fractures rates, prominent regional trends became apparent. As such, romosozumab resulted in a 42% reduction of non-vertebral fractures in rest-of-world (*p* = 0.012), whereas no treatment effect was observed in Latin America. The authors noted that background non-vertebral fracture risk was low in Latin America (1.2% in the placebo group), which could reflect a combination of genetic and environmental factors [78]. These findings imply that fracture risk assessment should consider regional factors in addition to the classical risk factors of BMD and fracture history. Accordingly, a sub-analysis of Japanese women in FRAME confirmed a similar efficacy of the romosozumab-to-denosumab regimen in these women compared to the overall FRAME population [79], while a sub-analysis of the East Asian cohort of ARCH showed that romosozumab followed by alendronate was associated with lower incidence of all fracture types studied (new vertebral, clinical, non-vertebral) vs. alendronate alone among East Asian patients [80].

In a prespecified exploratory study of FRAME, Geusens et al. analyzed the time-to-event for clinical vertebral fractures in the romosozumab and placebo arms within year 1 and reported a significant reduction in the incidence of clinical vertebral fractures through 12 months of romosozumab treatment, which only occurred within the first two months of therapy [81].

Lacking a study directly comparing romosozumab to denosumab efficacy, Cosman et al. retrospectively evaluated the BMD gains in FRAME and attempted to place these changes in the context of BMD increases seen in the Fracture Reduction Evaluation of Denosumab in Osteoporosis (FREEDOM) and its Extensions studies with denosumab alone [82]. The authors concluded that BMD improvements observed with romosozumab for one year in FRAME were comparable to those of 4.5 continuous denosumab treatment in FREEDOM and FREEDOM Extension [83]. Moreover, the two-year BMD gains at the lumbar spine and hip with one-year romosozumab followed by one-year denosumab approximated the effect of seven years of continuous denosumab administration [83]. This post-hoc analysis highlighted the advantages of a foundation effect with romosozumab before transitioning to an antiresorptive therapy.

In a post-hoc analysis of ARCH, Cosman and colleagues evaluated whether T-scores achieved at the total hip, femoral neck, and lumbar spine after one year of treatment with romosozumab or alendronate were related to subsequent risk of vertebral and non-vertebral fractures [84]. This study reported most robust relationships for the T-scores at the total hip, where T-scores achieved already at six months reflected future fracture risk, which could translate in the use of on-treatment total hip T-scores to monitor achievement of osteoporosis treatment goals [84].

A number of post-hoc sub-studies investigated the effects of romosozumab treatment on bone microarchitecture and bone stiffness. Graeff et al. assessed trabecular and cortical parameters via high-resolution quantitative computed tomography scans of the spine in 48 subjects of the multi-dose Phase I romosozumab study [66]. This analysis revealed significant increases in trabecular BMD, trabecular bone volume fraction and separation, and cortical thickness (all *p* < 0.05) through romosozumab treatment as compared to placebo. Moreover, bone stiffness, as estimated by finite element modeling, increased (+ 26.9%, *p* < 0.01 at month 3 compared to baseline). These improvements were maintained during the three months of follow-up [85]. Bone strength was assessed through simulated compression tests for the spine and through a sideways fall for the proximal femur with finite element analysis [86], in a sub-analysis of the international multi-dose Phase II romosozumab study [67]. Therein, at month 12, vertebral strength increased more for the highest romosozumab dose of 210 mg per month, compared to both teriparatide (27.3% versus 18.5%; *p* = 0.005) and placebo (27.3% versus −3.9%; *p* < 0.0001), while changes in femoral strength for romosozumab showed similar but smaller changes [86]. Interestingly, these changes in bone strength at the total hip closely correlated with changes in total hip areal BMD and bone mineral content, as measured by Dual-energy X-ray absorptiometry, and volumetric BMD, as measured by QCT in another sub-study [87]. A histological study performed on transiliac bone biopsies either after two or 12 months of treatment with 210 mg once monthly of romosozumab or placebo in a sub-study of the FRAME cohort confirmed an early and transient increase in bone formation, but a persistent decrease in bone resorption with romosozumab [88]. This resulted in an enhanced bone mass and trabecular thickness with improved trabecular connectivity, without significant changes of cortical porosity at month 12, as assessed by micro-CT [88].

The post-hoc analysis of STRUCTURE investigated the utility in early changes of P1NP in predicting BMD response [89]. Despite it being an attractive concept, this analysis indicated that in pre-treated patients with bisphosphonates, the P1NP increases after switching to an osteo-anabolic treatment do not help predict the hip BMD response [89].

#### 4.2.4. Extension Studies and Sequential Treatment Effects

The final analysis of FRAME through 36 months, where 12 months of romosozumab or placebo were followed by 24 months of denosumab treatment, has been published [90]. Of note, the magnitude of initial BMD improvement in romosozumab vs. placebo receivers during the first 12 months was maintained through 36 months, even though all patients were treated with denosumab for the last two years (10.5% at the lumbar spine, 5.2% at the total hip, and 4.8% at the femoral neck at 36 months for the romosozumab–denosumab group). Importantly, these BMD gains translated into a prolonged reduction of new vertebral and clinical fracture risk in prior romosozumab recipients. Thus, this study highlighted both the early and persistent fracture risk reduction benefits of romosozumab [90].

Various sequential regimens with romosozumab were explored in extension studies of the international multi-dose Phase II romosozumab study [67]. Data on BMD and bone turnover markers were reported on patients being treated with romosozumab for 24 months and subsequently being administered denosumab or placebo for 12 additional months [91]. Patients receiving romosozumab depicted BMD increases of 15.1% at the lumbar spine, 5.4% at the total hip, and 5.2% at the femoral neck (all *p* < 0.01 versus placebo). Of note, prior romosozumab recipients continued to accrue BMD between month 24 and 36 (i.e., with denosumab treatment) with additional mean gains of 2.6% at the lumbar spine, 1.9% at the total hip, and 1.4% at the femoral neck). Conversely, when romosozumab was followed by placebo, BMD decreased by 9.3% at the lumbar spine (remaining above baseline) and by 5.4% at the total hip (returning to pre-treatment levels) [91]. Bone turnover was, as expected, suppressed in denosumab recipients, while in patients transitioning to placebo, P1NP concentrations gradually returned to pre-treatment levels, and CTX levels increased and remained above baseline at month 36. Thus, this study showed that romosozumab does not show long-lasting effects after its discontinuation unless followed by antiresorptive treatment and highlighted the necessity of a subsequent therapy with an anti-remodeling drug.

After ascertaining the waning effect of romosozumab following its cessation, the next extension study evaluated the effect of a second course of romosozumab following a period off treatment or on denosumab [92]. Thus, participants of [91] were offered 12 additional months of 210 mg romosozumab per month (translating into months 36–48 of the original Phase II study [67]). In patients who received romosozumab at months 0–24 followed by placebo (months 24–36), a second romosozumab course increased BMD by comparable amounts to their first course at the lumbar spine (12.4% second course vs. 12% first course) and at the total hip (6.0% second course vs. 5.5% first course). By contrast, patients who were administered romosozumab at months 0–24 followed by denosumab (months 24–36) depicted a more modest BMD increase at the lumbar spine (2.3%) and maintained BMD at the total hip [92]. Thus, this study showed that the skeleton was fully reset to sclerostin inhibition after a period of 12 months without active therapy. Moreover, the sequence denosumab followed by romosozumab results in further (albeit more modest) BMD increases, which is an interesting observation, since a significant number of patients are pre-treated with antiresorptive agents before being switched to romosozumab in everyday clinical practice.

In the last stage of the extension studies described above [91,92], patients who had received active treatment for 48 months were assigned to no further active treatment, whereas all other women were administered a single intravenous dose of zoledronate 5 mg and evaluated for 24 months, up to month 72 [93]. In subjects receiving no further active treatment, a BMD decrease of 10.8% was noted at the lumbar spine from months 48–72, thereby remaining 4.2% above the original baseline. In patients assigned to zoledronate, BMD at the lumbar spine was maintained (percentage change of −0.8% between months 48–72 and 12.8% between months 0–72). Similar patterns were seen for BMD changes at the proximal femur in both groups. Regarding bone turnover, there was a decrease in P1NP and CTX concentrations, though remaining above baseline, at month 72, in patients receiving no further active treatment, while in zoledronate recipients, P1NP and CTX levels initially dropped and approached baseline by month 72 [93]. Thus, this study confirmed that romosozumab cessation resulted in bone mass loss, while a single zoledronate infusion preserved BMD for up to two years.

#### 4.2.5. Meta-Analyses

A number of meta-analyses with regard to romosozumab anti-fracture efficacy have been published to date. In a recent meta-analysis including only randomized controlled trials comparing different drugs for the management of postmenopausal osteoporosis, romosozumab depicted the lowest rate of new vertebral fractures [94]. A sophisticated network meta-analysis reported that while romosozumab, teriparatide, denosumab, and risedronate are the optimal treatments for secondary prevention of osteoporotic fractures in the setting of postmenopausal osteoporosis, only zoledronate has been shown to significantly reduce both vertebral and non-vertebral fractures for primary prevention in this setting [95]. Conversely, Ding and colleagues stratified various osteoporosis drugs in postmenopausal women either with or without prevalent vertebral fracture and observed that romosozumab was the only drug that could reduce clinical and vertebral fractures in both populations [96]. In a network meta-analysis comprising randomized controlled trials of osteoanabolic treatments, romosozumab, teriparatide, and abaloparatide significantly reduced the risk of vertebral fractures, but none of these drugs led to a significant reduction of the risk of non-vertebral fractures; in contrast, romosozumab consistently ranked better than teriparatide and abaloparatide with regard to BMD increase at all locations [97]. A preliminary meta-analysis of romosozumab regarding reduction of falls risk yielded a trend towards fewer falls after 12-months treatment with romosozumab [98].

## 5. Safety Profile of Romosozumab

As deficiency of sclerostin does not only result in high bone mass, but also leads to pathological features, the safety profile of romosozumab was carefully investigated.

In the first Phase I study, both subcutaneous and intravenous injections of romosozumab were assessed, with only one serious adverse event of nonspecific hepatitis, while no deaths or patients discontinuing the study were reported [65]. In the subsequent Phase I multiple-dosing study, romosozumab was administered subcutaneously, and no serious adverse events were reported. Additionally, side effects were comparable between the intervention groups and placebo [66]. In FRAME, adverse events were also balanced between groups, except for the very rare hypersensitivity reaction to romosozumab [69].

In the international Phase II study including 419 postmenopausal women, only mild generally non-recurring injection-site reactions were reported, with other adverse events similar between groups [67]. In the aforementioned network meta-analysis on postmenopausal women with or without a prevalent vertebral fracture, romosozumab was equally well tolerated as placebo, except for the rare hypersensitivity reactions [96], confirming the safety profile reported in FRAME [69].

Conversely, in ARCH, which included 4093 postmenopausal women assigned to either romosozumab or alendronate [71], a higher incidence of cardiovascular events was observed in patients treated with romosozumab than alendronate (0.8% vs. 0.3%) during the first year, especially with regard to a higher number of positively adjudicated serious cardiovascular events with romosozumab compared to alendronate (2.5 vs. 1.9% respectively) [71]. However, in the following 12 months during which patients in the romosozumab group were switched to alendronate, the rate of cardiovascular events did not change [71]. In men treated with romosozumab in the Phase III BRIDGE study, this increased cardiovascular risk was not observed; the numerical difference of positively adjudicated cardiovascular events in the romosozumab group was explained by a higher baseline risk with fewer subjects on cardioprotective medication [72]. Moreover, in the placebo-controlled FRAME study, there was no increased cardiovascular risk for romosozumab [69], nor was this the case in the study comparing romosozumab to teriparatide in subjects with prior bisphosphonate treatment [71]. In the latter study, side effects were even lower in the romosozumab group with the exception of atrial fibrillation [76]. A recent meta-analysis described an even lower cardiovascular risk in high-dose romosozumab treatment as compared to placebo [99]. In addition, a recent systematic review and meta-analysis concluded that the cardiovascular risk for elderly men and postmenopausal women with osteoporosis treated with romosozumab was not increased with regard to stroke, atrial fibrillation, heart failure, coronary artery disease, and cardiovascular death, although results were not completely conclusive [100] and a comparative trial is clearly needed. Taken together, the difference in rates of cardiovascular events between romosozumab and alendronate seen in ARCH could be attributable to a cardio-protective effect of alendronate, but further studies are necessary to conclusively address this issue [101].

The rare but severe complication osteonecrosis of the jaw, which has been described with antiresorptive treatment, has not been observed in preclinical studies with sclerostin inhibition [102], while two cases were reported in FRAME, but may be explained by dental procedures [69]. The meta-analysis considering all doses of romosozumab (210, 140, and 70 mg), and including studies with all comparators (alendronate, teriparatide, and placebo) reported a slightly increased risk of osteoarthritis, albeit not statistically significant [99]. Additionally, injection-site reactions were more likely with romosozumab treatment. In comparison to teriparatide, odds ratio for any adverse event was increased with romosozumab treatment [99]. A recent report described serious cardiovascular adverse events with romosozumab use in Japanese patients [103], although a sub-analysis from ARCH for East-Asian patients did not report an increased risk profile [80], and neither did the sub-analysis for Japanese patients from FRAME and its extension [79]. Mariscal et al. performed a meta-analysis in men and women treated with romosozumab and described that adverse effects of romosozumab were comparable to placebo, except for an increased risk of adverse reactions at the injection site [104]. An extensive network meta-analysis on primary and secondary prevention of osteoporotic fractures demonstrated a tolerability and acceptability similar between romosozumab, teriparatide, and denosumab compared to placebo [95]. In summary, romosozumab appears to be a safe and well-tolerated drug. However, primarily because of the findings of the ARCH study [71], the FDA approved romosozumab with a boxed warning that it may increase the risk of myocardial infarction, stroke, and cardiovascular death. It should not be used in patients with a myocardial infarction or stroke in the year previous to therapy or while on therapy, and the benefits and risks should be carefully outweighed in patients at high risk of cardiovascular events.

## 6. Translational Applications of Sclerostin Antibodies in Diseases other than Osteoporosis

### 6.1. Osteogenesis Imperfecta (OI)

In OI type III mice, Scl-Ab treatment reduced the fracture rate of long bones by improving bone mass, density, and biomechanical strength [105,106,107]. A significant reduction in vertebral fractures was also reported [108], although these effects could not be replicated in a more severe murine phenotype of OI [109]. In adults with OI, treatment with Scl-Ab resulted in an increased bone formation with a reduction in bone resorption and a subsequent rise in BMD [110]. Because of the known waning effect of Scl-Ab after treatment cessation [64], anti-resorptive treatment must be considered afterwards. As such, bone loss could be overcome by a single intravenous injection of pamidronate in a preclinical model of OI [111]. Moreover, OI animal models show that low dose bisphosphonate treatment amplified the effect of Scl-Ab treatment during the early stage of skeletal growth [112,113]. A study to evaluate the pharmacokinetics profile following multiple subcutaneous doses of romosozumab in children and adolescents with OI will start recruiting in 2021 (NCT04545554).

### 6.2. X-Linked Hypophosphatemia (XLH)

X-linked hypophosphatemia (XLH) is a form of rickets with low phosphate levels due to renal loss with increased fibroblast-growth factor-23 levels. In XLH mice, in which sclerostin levels are high, treatment with Scl-Ab resulted in increased phosphate levels and suppressed circulating fibroblast-growth factor-23 levels, findings which coincided with increased bone volume and peak load [114] as well as mineralization [115]. Thus, these preclinical data suggest that Scl-Ab may also be effective in the treatment of XLH.

### 6.3. Malignant Disease

To date, Scl-Ab treatment has been proven safe with regard to malignancy occurrence. In particular, it has been shown that although treatment with Scl-Ab in multiple myeloma mouse models did not inhibit bone resorption and did not reduce tumor burden, it prevented myeloma-induced bone loss, reduced lytic bone lesions, and increased fracture resistance [116]. Moreover, since breast cancer cells secrete sclerostin [117], they inhibit osteoblast differentiation and thus bone formation. Treatment of skeletally metastasized mice with Scl-Ab alleviated growth of bone tumor lesions and bone destruction. It also decreased muscle atrophy and prolonged life of the mice [117]. Thus, Scl-Ab appears to have significant potential for treatment of malignant skeletal metastasized diseases.

### 6.4. Other

Due to their disease modifying effects, Scl-Ab could prove efficacious as treatment for bone pain relief [118] and periodontal disease [119,120,121]. In patients with hypophosphatasia, an increment in bone formation biomarkers and BMD was observed with Scl-Ab treatment [122], suggesting a possible role for Scl-Ab treatment in this disease.

In a mouse model of osteoporosis pseudoglioma syndrome, depletion of sclerostin had an anabolic effect [123], suggesting that these patients might benefit from Scl-Ab therapy.

Treatment with Scl-Ab in rats with renal osteodystrophy did not show efficacy of Scl-Ab unless PTH levels were low [124] or medically suppressed by calcium application [125].

Scl-Ab treatment did not prevent growth retardation in young mice treated with dexamethasone [126]. Scl-Ab improved axial bone mass in mice with rheumatoid arthritis, but did not affect systemic inflammation, nor did it prevent local erosions [127]. In fact, sclerostin inhibition worsened TNF-dependent joint destruction [128]. Scl-Ab inhibits loss of bone mass in mice exposed to partial weight-bearing [129], thereby posing Scl-Ab as an interesting drug in the prevention or treatment of bone loss in astronauts.

In rodents, Scl-Ab did not prevent muscle atrophy after spinal cord injury [130]. However, as patients with non-ambulatory cerebral palsy have higher sclerostin levels [131], suggesting it may be a therapeutic target, a study to determine the effects of monthly romosozumab for one year or one-time zoledronic acid on BMD and bone turnover markers in patients with spinal cord injury and low BMD will start recruiting in 2021 (NCT04597931).

Two recent studies investigated the potential of romosozumab to accelarate fracture healing [132,133]. The effect of three doses of romosozumab (70, 140, or 210 mg) or placebo administered postoperatively on day 1 and weeks 2, 6, and 12 on open reduction and internal fixation of intertrochanteric or femoral neck hip fractures was investigated [132]. This study did not reveal any improvement of radiographic or functional clinical outcomes through romosozumab with regard to acceleration of fracture healing [132]. Similar results were reported in a study of surgical fixation of tibial diaphyseal fractures, where nine different regimens of romosozumab or placebo were implemented without radiographic or clinical benefit in fracture healing [133].

## 7. Conclusions

The inhibition of sclerostin, due to its dual mode of action on enhancing bone formation and simultaneously suppressing bone resorption, is one of the most potent means of increasing BMD to date. With the clinical development of romosozumab, a safe and efficacious drug is available to treat osteoporosis in postmenopausal women and men. As its potency after repeated injections wanes, follow-up strategies will be necessary to maintain the high increase in BMD. Additonally, future studies will show for which other diseases inhibition of sclerostin may be a useful treatment opportunity.

## Figures and Tables

**Figure 1 jcm-10-00787-f001:**
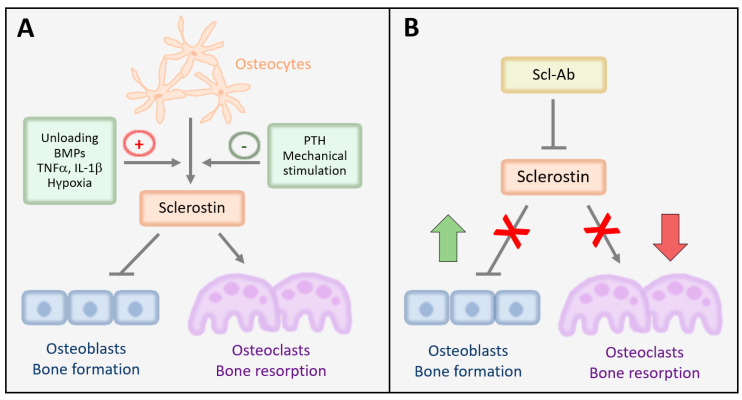
Regulation of sclerostin and its effects on bone cells and mode of action of sclerostin antibodies. (**A**) Regulation of sclerostin and its effects on bone cells. Sclerostin is produced mainly by osteocytes and is negatively regulated by mechanical loading and parathyroid hormone (PTH) and positively by unloading, bone morphogenetic proteins (BMPs), pro-inflammatory cytokines (TNFα, IL-1β), and hypoxia. Sclerostin inhibits osteoblastogenesis and thus bone formation and stimulates osteoclastogenesis and bone resorption. (**B**) Mode of action of sclerostin antibodies. Sclerostin-antibodies (Scl-Ab) neutralize sclerostin and thereby stimulate bone formation and inhibit bone resorption, producing a large anabolic window.

## Data Availability

Not applicable.

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
