# Peer review of "Osteoporosis Treatment with Anti-Sclerostin Antibodies—Mechanisms of Action and Clinical Application"

_jcm, 2021, doi:10.3390/jcm10040787_

Round 1
Reviewer 1 Report
The current manuscript by Rauner et al is a comprehensive review of literature on anti-sclerostin antibodies, and their impact on Osteoporosis. The authors discusses the role of sclerostin in bone physiology and osteoporosis progression. They describe at length the cellular mechanisms and models studying sclerotin and it's inhibition. The review reveals the major implications of pharmacological inhibition of sclerostin and associated clinical trials and meta-analyses published. Overall this is a comprehensive and well written review that provides the reader with a strong and clear understanding of the role of sclerostin inhibition in bone formation and resorption and highlights the areas of further study. I recommend the manuscript be accepted as is.
Author Response
We thank the reviewer for her/his positive evaluation.
Reviewer 2 Report
The review provides a comprhensive overview about the clinical investigation romosozumab. The study related to the clinical development are well presented and explained for the readers
In my opinion paper is quite suitable for pubblication.
Author Response
We thank this reviewer for her/his positive evaluation.
Reviewer 3 Report
Title: Osteoporosis treatment with anti-sclerostin antibodies – mechanisms of action and clinical application
ID: JCM-1086620
Authors: Martina Rauner, Hanna Taipaleenmäki, Elena Tsourdi and Elizabeth M. Winter
General Comments: In this review, the authors talk about osteoporosis, sclerostin, its mechanism of action as well as the effects of sclerostin knockdown in preclinical models. In depth analysis of anti-sclerostin antibody treatment options available as well as the outcomes of various clinical studies carried out are also laid out in detail. The tabulated summary of romosozumab trials lays out the details in a very concise and comprehensible way. Aside from discussing the important findings from the clinical studies, the authors also talk about the information gleaned from post-hoc analyses of existing clinical data. The applications of anti-sclerostin antibodies to other skeletal diseases is a useful section in gaining a more thorough understanding of the advantages of this treatment modality. Some minor modifications may be beneficial to the manuscript as detailed below in the specific comments section.
Specific Comments:
- Please mention abbreviations alongside the full name of the gene prior to using it solo in the text like SOST is mentioned on page 2.
- I would suggest to perhaps reorder the sections i.e. explaining what SOST is first before delving into its role in regulating bone mass.
- Please be consistent and use the correct formatting (upper case or lower case letters; italicized or not) for gene and protein names based on the studies being cited.
- There is no description of denosumab prior to its mention in the romosozumab section. Please write a line or two about it for the benefit of readers unfamiliar with sclerostin antibodies.
Author Response
We thank this reviewer for her/his careful evaluation.
Specific Comments:
- Please mention abbreviations alongside the full name of the gene prior to using it solo in the text like SOST is mentioned on page 2.
We have mentioned abbreviations alongside the full name of the genes.
- I would suggest to perhaps reorder the sections i.e. explaining what SOST is first before delving into its role in regulating bone mass.
We have reordered the sections according to the reviewer’s suggestion.
- Please be consistent and use the correct formatting (upper case or lower case letters; italicized or not) for gene and protein names based on the studies being cited.
We have unified the presentation of mouse/human gene names throughout the manuscript.
- There is no description of denosumab prior to its mention in the romosozumab section. Please write a line or two about it for the benefit of readers unfamiliar with sclerostin antibodies.
We have added a sentence about denosumab and a respective reference on pages 13-14.
Reviewer 4 Report
The manuscript is well-written and fluent in all its parts. The authors applied for many research aspects related to bone metabolism. The topic is very relevant but needs some additional minor revision prior to publication.
This is a comprehensive review regarding the pre-clinical and clinical studies regarding a new anabolic drug - romosozumab - for the treatment of osteoporosis. The authors should highlight the fact that it is an opinion-based and not a systematic review throughout the entire manuscript, including the abstract section.
Minor points:
INTRODUCTION: One of the effective measures to prevent fractures is to increase adherence to available treatments, i.e. denosumab. The authors are encouraged to quote Betella N et al. doi: 10.23736/S0391-1977.20.03137-5 and Migliaccio S et al. doi: 10.1007/s40618-017-0701-3
SCLEROSTIN:
The authors should consider the preclinical evidence that aspirin may exert additional protective effects on OPG-RANKL-RANK signaling pathway and quote doi: 10.23736/S0391-1977.
SAFETY:
Some additional comment on “black box warning” present in the data sheet of this drug, should be better expanded and patients not candidate to use this drug should be explicited.
OTHERS:
The authors should highlight indication to treatment with romosozumab in particular male subpopulations affected by endocrine disease i.e. primary aldosteronism (doi: 10.23736/S0391-1977.18.02867-5), hypogonadism (doi: 10.23736/S0391-1977.20.03195-8) or diabetes (doi: 10.1007/s12020-017-1480-5), where the indication to the use this drug has not been tested and its use maybe be additional to the treatment of the primary condition.
Author Response
We thank this reviewer for her/his careful evaluation and have added a sentence in the abstract and introduction that this is an opinion-based narrative review and not a systematic review.
Minor points:
INTRODUCTION: One of the effective measures to prevent fractures is to increase adherence to available treatments, i.e. denosumab. The authors are encouraged to quote Betella N et al. doi: 10.23736/S0391-1977.20.03137-5 and Migliaccio S et al. doi: 10.1007/s40618-017-0701-3
We have added a sentence about adherence and have included these two references.
SCLEROSTIN:
The authors should consider the preclinical evidence that aspirin may exert additional protective effects on OPG-RANKL-RANK signaling pathway and quote doi: 10.23736/S0391-1977.
As we believe that the interactions of aspirin with the RANK-RANKL-OPG signaling pathway are a bit off topic for our review on romosozumab, we have not added this information.
SAFETY:
Some additional comment on “black box warning” present in the data sheet of this drug, should be better expanded and patients not candidate to use this drug should be explicited.
We have added the box warning on cardiovascular events. We stressed the need to outweigh risks and benefits in patients at high risk for myocardial infarction, and that it should not be prescribed in patients with a recent myocardial infarction or stroke.
OTHERS:
The authors should highlight indication to treatment with romosozumab in particular male subpopulations affected by endocrine disease i.e. primary aldosteronism (doi: 10.23736/S0391-1977.18.02867-5), hypogonadism (doi: 10.23736/S0391-1977.20.03195-8) or diabetes (doi: 10.1007/s12020-017-1480-5), where the indication to the use this drug has not been tested and its use maybe be additional to the treatment of the primary condition.
We agree with the reviewer that the efficacy of romosozumab should be validated in the men with osteoporosis, especially in the setting of secondary male osteoporosis, and have added this information and relevant references on page 15.